# Early-, Late-, and Very Late-Term Prediction of Target Lesion Failure in Coronary Artery Stent Patients: An International Multi-Site Study

**Elisabeth Pachl** [1,†] ⓘ, **Alireza Zamanian** [1,†], **Myriam Stieler** [2], **Calvin Bahr** [2] and **Narges Ahmidi** [1,3,*] ⓘ

1   Helmholtz Munich Center, Computational Health Department, 85764 Munich, Germany; elisabeth.pachl@helmholtz-muenchen.de (E.P.); alireza.zamanian@helmholtz-muenchen.de (A.Z.)

2   Medical Affairs Department, Biotronik AG, 8180 Bülach, Switzerland; myriam.stieler@biotronik.com (M.S.); calvin.bahr@biotronik.com (C.B.)

3   Fraunhofer Institute for Cognitive Systems, 80686 Munich, Germany

\*   Correspondence: narges.ahmidi@helmholtz-muenchen.de

†   These authors contributed equally to this work.

**Abstract:** The main intervention for coronary artery disease is stent implantation. We aim to predict post-intervention target lesion failure (TLF) months before its onset, an extremely challenging task in clinics. This post-intervention decision support tool helps physicians to identify at-risk patients much earlier and to inform their follow-up care. We developed a novel machine-learning model with three components: a TLF predictor at discharge via a combination of nine conventional models and a super-learner, a risk score predictor for time-to-TLF, and an update function to manage the size of the at-risk cohort. We collected data in a prospective study from 120 medical centers in over 25 countries. All 1975 patients were enrolled during Phase I (2016–2020) and were followed up for five years post-intervention. During Phase I, 151 patients (7.6%) developed TLF, which we used for training. Additionally, 12 patients developed TLF after Phase I (right-censored). Our algorithm successfully classifies 1635 patients as not at risk (TNR = 90.23%) and predicts TLF for 86 patients (TPR = 52.76%), outperforming its training by identifying 33% of the right-censored patients. We also compare our model against five state of the art models, outperforming them all. Our prediction tool is able to optimize for both achieving higher sensitivity and maintaining a reasonable size for the at-risk cohort over time.

**Keywords:** machine learning; TLF prediction; multi-site clinical cohort; stent intervention

## 1. Introduction

Cardiovascular disease (including coronary artery disease) is regarded as the most severe and lethal disease in humans. According to data provided by the WHO, it causes one third of all deaths globally (approximately 17.9 million p.a.) [1]. In coronary artery disease, the arteries of the heart are blocked or narrowed from a buildup of cholesterol and fatty substances. This obstruction reduces the blood flow to the heart or forms a blood clot, leading to a heart attack. Existing guidelines [2] recommend coronary stents as the default means of treatment. Stent interventions are routine procedures; the stent is inserted into the affected vessel, expanded, and left inside the body permanently. More than four million stents are inserted annually worldwide [3] (incl. 29,000 in Switzerland, 25,000 in Austria, and 379,000 in Germany [4]).

There are guidelines and risk score models for the prediction of general cardiac events for stable or at-risk patients. These models use a variety of features such as clinical parameters, computed tomography (CT) and magnetic resonance imaging (MRI), and radiomics [5–7] to segment the structure [6,8] of the cardiovascular system, extract quantitative features from medical images (Radiomics), diagnose the potential risk, and predict a future adverse event (e.g. myocardial infarction). These models focus on understanding

the risk factors for relatively stable populations, before they require surgical treatment. In contrast, in our study, we focus on the development of adverse events after patients receive stent implantation treatment. Figure 1 highlights the focus of our investigation and how it complements existing guidelines.

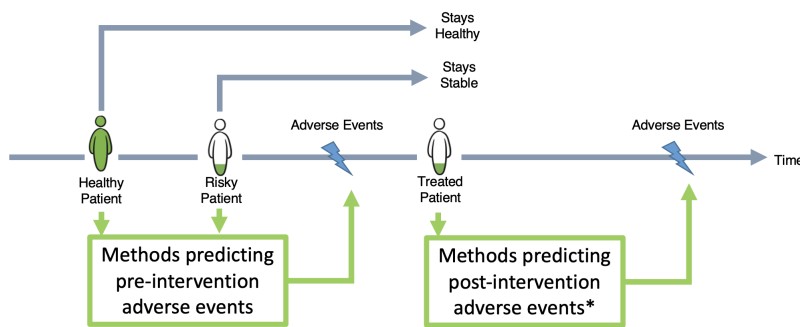

**Figure 1.** Landscape of different methods in the literature predicting adverse cardiac events. Some methods are designed to predict these events for stable or healthy populations, while others focus on sick and already treated patients. The focus of our study (*) is to predict post-intervention target lesion failure. TLF is defined as the composite of multiple clinical endpoints: cardiovascular death, clinically driven target lesion revascularization, target vessel myocardial infarction, and stent thrombosis.

In the remainder of the paper, the outcome of interest is post-intervention target lesion failure (TLF). TLF is defined as a composite of multiple clinical endpoints: cardiovascular death, clinically driven target lesion revascularization, and target vessel myocardial infarction [9]. Another clinically relevant endpoint is stent/scaffold thrombosis. For a better overview and readability, scaffold thrombosis are also be assigned to the TLF category throughout this paper.

In this study, we aim to predict TLF as an adverse event after a stent surgery by using a novel ML approach and an international cohort. This may allow physicians to identify at-risk patients earlier, and to prioritize resources for follow-up care.

Our patient data were collected in the Biotronik BIOSOLVE-IV post-market study (2016–2025) [10], a prospective, single-arm, multi-center, open-label registry which aims to investigate the clinical performance and long-term safety of the bioresorbable stent Magmaris in a real-world setting. The study has been approved by the relevant ethics committees and competent authorities, and all patients have provided written informed consent.

The five main challenges for building an early prediction system for TLF are: (1) there are no clinically accepted bio-markers (features) for identification of at-risk patients. (2) As a decision support tool, it is important to optimize between the tool's sensitivity and the size of the at-risk cohort. (3) Model training is hampered by unbalanced class sizes (only 7.6% patients experienced TLF). (4) Missingness in the observations (nested input observations or right-censoring of the outcome) needs to be handled appropriately to avoid bias in inference. (5) Clinical studies usually suffer from selection bias due to restrictive cohort inclusion criteria.

Our main contributions in this paper are as follows:

1. The development of a novel ML method for the identification of patients at risk of TLF. This model can predict the onset of TLF at any time point after discharge from hospital. To the best of our knowledge, such an analysis on TLF is the first of its kind.
2. Training and validation are performed with our international, multi-site cohort, with comprehensive variables collected during treatment and 5 years of frequent follow-ups. Data were collected from 120 medical centers in over 25 countries across the globe.
3. In addition, we evaluate our model against five state of the art models via multiple sets of experiments.

4. We demonstrate a successful retrospective and prospective evaluation of our model in three time frames (early-, late-, and very late-term prediction).

In Section 2, we first summarize the state of the art in the field of risk prediction and modeling of adverse events after stent surgery. In Section 3, we introduce the cohort and data collected for this study. In Section 4, we provide details of our methodology pipeline. In Section 5, we evaluate our model and compare it with the state of the art techniques for prediction of TLF. Finally, Section 6 summarizes our findings, strengths and weaknesses, and suggests guidelines for further improvements.

## 2. Background

Early prediction of TLF via analysis of large cohorts can help improve patient outcomes and optimize their follow-up care. Tables 1 and S1 summarize the state of the art in this field. The first risk prediction methods for coronary artery disease were developed in 2004 based on multivariate regression [11]. More recently, guidelines were established for the prediction of adverse events for coronary artery disease patients, such as the Framingham Risk Score in 2010 [12] or the Systematic Coronary Risk Evaluation algorithm in 2016 [13]. Apart from these, there also exist statistical methods on possible risk factors for TLF for various stent technologies [14,15]. Those algorithms are based on multivariate regression incorporating well-known risk factors, such as diabetes, smoking habit, and age, alongside their in-between interactions available from prior knowledge. In general, they exhibit only modest predictive power due to relatively strong model assumptions.

A few existing machine learning (ML) approaches have been applied [16,17] for TLF prediction to date. The benefit of ML methods compared to previously used methods is that they can search large amounts of data and identify multi-variable patterns that reliably predict outcomes of interest [18,19]. However, the existing studies for TLF prediction suffer from the following issues: (1) there exist only simple state models of hand-picked variables, (2) the sizes of the cohorts are limited and they are mostly from single-site hospitals, (3) comprehensive observations about patients are lacking, and (4) the observations are rarely longitudinal, without longer-term follow-up.

In this paper, we aim to identify the post-intervention TLF outcome, months before its onset, by using a novel ML approach and an international cohort. This may allow physicians to identify at-risk patients much earlier and to prioritize their follow-up care.

We compare our model to five state of the art models: (1) PRESTO-1 [20], (2) PRESTO-2 [11] both conducted by Sing et al. in 2004, (3) EVENT [21] conducted by Stolker et al. in 2010, (4) Konigstein [15] conducted by Konigstein et al. in 2019, and (5) GRACIA-3 [17] conducted by Sampedro-Gómez et al. in 2020. The first three models are used extensively in clinical practice. They were originally designed to predict the risk of developing stent restenosis (PRESTO-1, PRESTO-2) or TLR (EVENT) for post-intervention patients.

The fourth model, Konigstein, was originally designed to predict post-intervention TLF using a univariate Cox proportional hazard model, and the fifth is a recent ML-based model trained on the GRACIA-3 cohort for the prediction of stent restenosis. All these models have a large overlap in their features with our database and involve similar cohorts, and hence are included in our comparison. We excluded other state of the art methods in our comparison because of the following two reasons: (1) there [5,14,22] was little to no overlap between their respective feature sets and ours; for example, these methods are mainly based on features extracted from CT images or angiography. Such features could not be routinely collected in follow-ups of our multi-national cohort, not least due to resource disparities between the 25 participating countries. (2) They were designed to study cardiac risk factors in a rather stable or healthy population [5,12,16], leading to the exclusion of patients with previous histories of stent intervention, who, however, comprise the entirety of our cohort. Therefore, a head-to-head comparison is not meaningful.

**Table 1.** Summary of literature review. CAD = coronary artery disease, CVD = cardiovascular disease, SR = stent restenosis, TLR = target lesion revascularization, SCL = supervised classification.

| Paper | Outcome | Data | | Method | Metric |
|---|---|---|---|---|---|
| Singh, 2004 (PRESTO-1, PRESTO-2) [11,20] | SR | Cohort name | PRESTO trial | Multi-class LR | AUC-ROC |
| | | # Patients | 1312 | | |
| | | Ethnicity | Heterogeneous worldwide, 224 hospitals | | |
| | | Outcome ratio | 45.4% | | |
| | | Medical device | BMS | | |
| | | Features | Baseline demographic, procedural, and follow-up angiographic information | | |
| D'Agostino Sr, 2008 (Framingham Risk Score) [12] | CAD | Cohort name | Framingham study | Cox proportional-hazards regression | C-index |
| | | # Patients | 8491 | | |
| | | Ethnicity | Participants from one city (Framingham, Massachusetts) | | |
| | | Features | Baseline demographic and socioeconomic, CAD phenotype (incl. genetic biomarkers), procedural, and follow-up angiographic information | | |
| Stolker, 2010 (EVENT) [21] | TLR | Cohort name | EVENT registry | Multi-class LR | C-index |
| | | # Patients | 5863 | | |
| | | Ethnicity | Heterogeneous from USA | | |
| | | Outcome ratio | 4.1% | | |
| | | Medical device | DES | | |
| | | Features | Demographic, clinical, and treatment features | | |
| Cassese, 2014 [22] | SR | # Patients | 10,004 | Fisher's exact test, $X^2$ test | *p*-value |
| | | Ethnicity | Homogeneous German, two hospitals | | |
| | | Outcome ratio | 26.4% | | |
| | | Medical device | BMS or 1st/2nd-generation DES | | |
| | | Features | Baseline demographic, procedural, and follow-up angiographic information | | |
| Alaa, 2019 [16] | CAD | Cohort name | UK Biobank | SCL | AUC-ROC |
| | | # Patients | 423,604 | | |
| | | Ethnicity | Homogeneous from UK, 22 hospitals | | |
| | | Outcome ratio | 1.1% | | |
| | | Features | Baseline demographics, procedural, and follow-up angiographic information | | |
| Konigstein, 2019 [15] | TLF | Cohort name | Pool from six randomized controlled trials | Cox proportional-hazards regression | C-index |
| | | # Patients | 10,072 | | |
| | | Ethnicity | Heterogeneous, worldwide | | |
| | | Outcome ratio | 10.1% | | |
| | | Medical device | Contemporary DES | | |
| | | Features | Baseline demographic, procedural, and follow-up angiographic information | | |
| Anadol, 2020 [14] | TLF | Cohort name | MICAT project | Cox proportional-hazards regression | C-index |
| | | # Patients | 512 | | |
| | | Ethnicity | Heterogeneous, worldwide (14 countries) | | |
| | | Outcome ratio | 17.9% | | |
| | | Medical device | BRS (Abbott Vascular) | | |
| | | Features | Baseline demographic, procedural, and follow-up angiographic information | | |
| Sampedo-Gómez, 2020 [17] | SR | Cohort name | GRACIA-3 study | SCL | AUC-PRC |
| | | # Patients | 263 | | |
| | | Ethnicity | Homogeneous Spanish | | |
| | | Outcome ratio | 8.9% | | |
| | | Medical device | DES | | |
| | | Features | Baseline demographic, procedural, and follow-up angiographic information | | |

### 3. Data and Cohort

Second-generation drug-eluting stents are considered gold-standard devices in stent surgery, with an overall TLF rate of 7% [23]. Bioresorbable stent technology is the fourth revolution in interventional cardiology [24], and is designed to address medical shortcomings of drug-eluting stents; their purpose is to fully support the vessel during the critical period, and then be resorbed by the body when no longer needed. The TLF rate of first-generation bioresorbable stents (with a poly-L-lactic acid polymer backbone, e.g., ABSORB bioresorbable vascular scaffolds (Abbott Vascular) [25]) is reported to be about 8–11% [26]. For second-generation bioresorbable stents (with a magnesium alloy backbone, e.g., Magmaris sirolimus-eluting bioresorbable stents (Biotronik AG, Bülach, Switzerland)), the TLF rate is reported to be about 3.3–6.8% [27].

The Magmaris stent consists of a limus family drug and a bioabsorbable polymer matrix, which achieves a controlled drug release for up to 90 days [28]. They are available in diameter ranges of 3.0–3.5 mm and lengths of 15 mm, 20 mm, and 25 mm. Pre- and post-dilatation are mandatory, for which a non-compliant balloon with a 1:1 balloon-to-artery ratio is used. The residual stenosis after pre-dilatation should not exceed 20.0% [29].

Cohort: In the BIOSOLVE-IV prospective study, we collect data from Magmaris implantations in adult patients (age > 18) between September 2016 and 2025 (last enrolled in July 2020). Each patient is followed up for up to five years. This study is being conducted in more than 120 medical centers in over 25 countries in Europe, Asia, Africa, and Australia/New Zealand [30]. The main inclusion criteria are target lesion stenosis >50.0% and <100.0%, thrombolysis in myocardial infarction (TIMI) flow $\geq 1$, and a reference vessel diameter of 2.7–3.7 mm by visual estimation. However, the study also includes complex lesions except for occlusions. The main exclusion criteria are pregnancy, dialysis, left main coronary artery disease, restenotic lesions, or acute ST-elevated myocardial infarction [30]. Figure S1 shows an overview of the BIOSOLVE-IV prospective study.

Data Elements: In the supplementary document, Tables S2–S7 show detailed statistical descriptions of the 86 data elements and feature characteristics for the cohort used for this analysis. Here in the main manuscript, Table 2 shows a summary of those features, following the timeline of the data collection per individual. The pre-intervention data (Table S2) include demographics, physical examinations, medical history, risk factors, and questionnaires. All events during the intervention are reported in Table S3. In addition, lesion characteristics and the Magmaris implantation itself are documented very closely (Table S4). Between the end of the procedure and hospital discharge, concomitant medication (Table S5), in accordance with the hospital standard of care, laboratory values, and ischemic status (Table S6) are also documented. After the procedure, all patients undergo follow-up evaluations at six months, twelve months, and annually after that, for up to five years. At each follow-up, data on the status of ischemia/angina, adverse events, and questionnaires are collected. Most of the features show an overall low missingness rate (Figure S2).

Until July 2020, there were 2066 patients enrolled in the BIOSOLVE-IV study. We excluded patients with a missingness rate higher than 50% (*n* = 3) (Figure S2) and patients treated with more than one Magmaris stent (*n* = 85). We included all TLF patients. The remaining 1975 patients were used for analysis. In the cohort (*n* = 1975), there is a total of 163 patients with TLF (8.25%): 151 patients had TLF before end of Phase I, and 12 afterwards, and 84.14% of all TLF events happened within the first year after implantation. For patients with multiple TLF events (*n* = 4), we only considered the time of their first event.

**Table 2.** Summary of 86 features used in our study, grouped by the timeline of treatment for each individual patient.

| Timeline | Supplementary References | Features |
|---|---|---|
| Pre-intervention | Table S2 | Demographic information (age, gender), EQ5D questionnaire (mobility, self-care, usual activities, pain/discomfort, anxiety/depression), MI information (prior MI, type of most recent MI), prior stroke/TIA, diseases and complications (renal, hepatic, respiratory, hypertension, hypercholesteremia, diabetes mellitus, congestive heart failure), history of cancer, number of prior PCIs, ischemic status (STEMI, NSTEMI, CCS class of stable angina, unstable angina, silent ischemia, LVEF class) |
| Intra-operation | Table S3 | Procedure details (duration, residual stenosis before implantation), device (Magmaris scaffold) details (number of implanted devices, maximum pressure applied, residual stenosis after implantation, device deficiency prior to/during procedure), pre-dilatation balloon details (number of balloons, diameter, length, number of inflations, maximum pressure), post-dilatation balloon details (same as for pre-dilatation) |
| Lesion and Stent | Table S4 | Lesion information (location, ACC/AHA characterization, moderate/severe calcification, eccentric lesions, length), vessel information (location, moderate/severe angulation, moderate/excessive tortuosity, reference diameter), pre-procedure TIMI flow, bifurcation, thrombus, stenosis pre-procedure |
| Medications | Table S5 | ASA (prior to procedure, loading dose), heparin (bolus injection prior to procedure, during procedure), anti-platelet medication (prior to procedure, loading dose) |
| Discharge Information | Table S6 | Troponin (if clinically significant, if out of normal range), ischemic status (CCS class of stable angina, unstable angina, silent ischemia) |
| Follow-up | Figure S1 | TLF defined as combination of TLR, MI, CABG, and ST |

## 4. Methodology

We developed an ML model with three main prediction components: (1) the first component aims to predict the TLF outcome at discharge, (2) the second component aims to predict a continuous score for time to TLF event, and (3) the third component updates the risk score for patients during their follow-ups. Figure 2 shows the algorithm flow.

### 4.1. TLF Prediction Component

The TLF prediction model component encompasses nine ML prediction models along with the pre-processing steps required for resolving missingness, adjusting sample size, reducing dimensionality, and normalizing the data, as well as a complete nested cross-validation setup. The pre-processing steps are designed to be optionally turned on and off during the model selection phase.

A large part of the inconsistency and lack of produciblity in the ML literature comes from hidden pre-processing steps. In the following section, we explain how incorrect execution of these pre-processing steps introduces bias in the final results and findings of any ML models.

(1) Transformation: To correct for differences in feature scales and outliers, we employ a scaling step in order to prevent convergence issues and unreliable model training. Since the straightforward normalization cannot be applied to missing data, the Quantile Transformer can be considered a possible alternative. First, it maps the feature values to a uniform distribution using the feature's empirical cumulative distribution function (eCDF), and then maps the transformed values to a target distribution by applying the associated quantile function. This operation is monotone and rank-preserving. The main benefit of this technique is alleviating the effect of outliers by scaling them into a sensible

range and dispersing extremely close values which will potentially lead to better model training. However, this means that the in-between sample correlations do not remain the same, of which we should be aware.

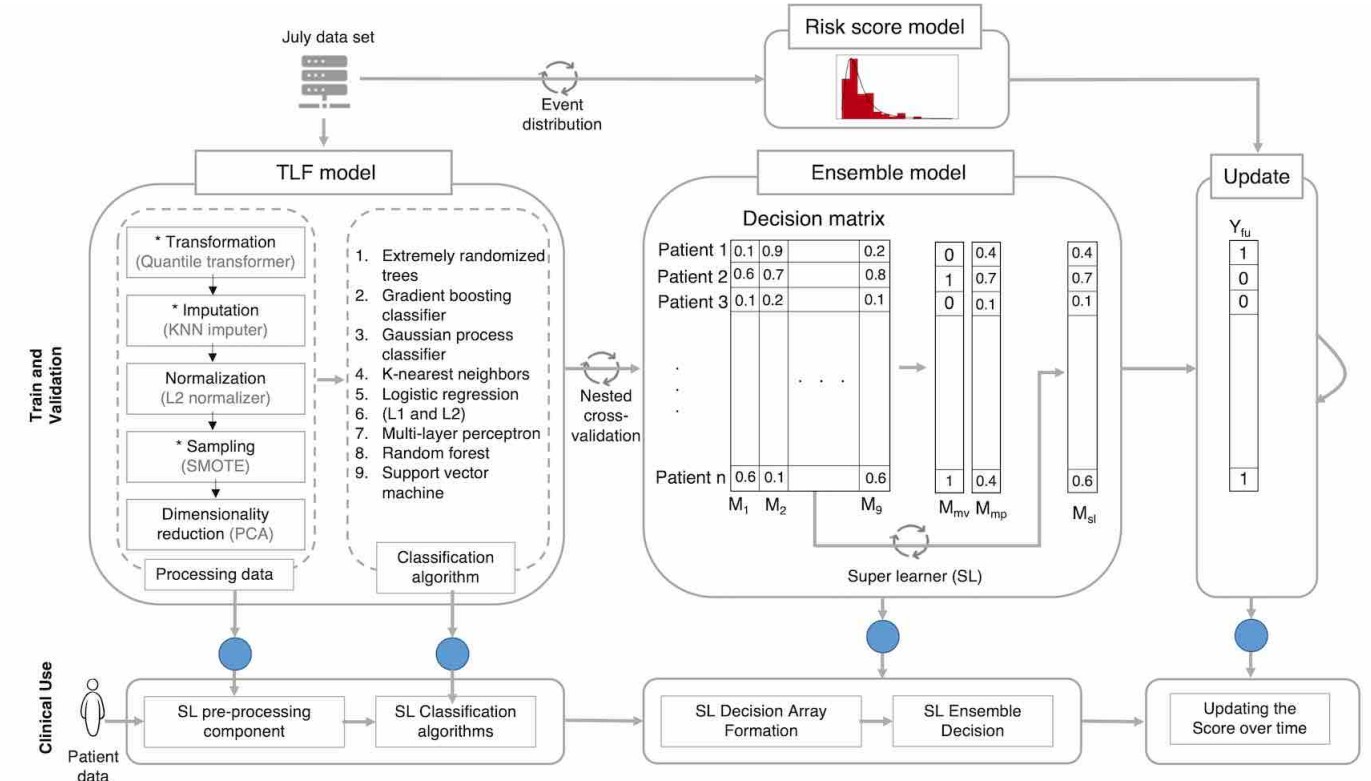

**Figure 2.** Model architecture and its main components for when the model is being trained and validated (**top**), and when in use in clinics (**bottom**). The model deployed in clinics requires minimal computation power and resources, since it only requires the final trained parameters of the system (blue circles). Only at the beginning do we require above-average computation power to train and validate the models.

(2) Imputation: We performed imputation to deal with missingness in the dataset. We opted to use conventional imputation over more stable approaches such as inverse-probability weighting [31] as no particular parameter was of higher interest a priori and the data were to be used as input for various downstream classifiers. Therefore, we used the non-parametric model-based k-nearest neighbor (KNN) imputation technique [32]. It does not require explicit missingness modeling and is a valid imputation technique for the missing-at-random assumption of the data generation mechanism [31]. Using more robust techniques such as MICE [33] or MissForest [34] did not show any significant improvement over the simpler and faster KNN imputer. The main hyper-parameter of the KNN imputer is $K$, the number of neighbors in the KNN algorithm. We used Euclidean distance and weighted averaging for similarity measurements and voting, respectively.

(3) Normalization: Similar to the Quantile Transformer, this step also provides an option to scale the data, but after imputation, since it cannot handle missing data, as opposed to the Quantile Transformer. We opted for the L2-norm for normalization.

(4) Sampling: Many ML algorithms are prone to under-fitting, especially when class sizes are imbalanced, i.e., one class (no TLF class in our case) has predominantly populated the dataset. To minimize the model loss function, classifiers then usually overrepresent the dominant class and ignore the minority class(es). A general approach is thus to undersample the dominant class or oversample the minority class. The synthetic minority oversampling technique (SMOTE) is one of the most commonly used oversampling algorithms. It generates virtual training samples by randomly spaced linear interpolation for the minority class:

$$x' = x + \text{gap}|x - x_k|, \quad \text{gap} \sim \text{Unif}(0, 1)$$

where $x'$ is the new training sample in the minority class, generated from sample $x$ in the same class and random combination (uniform distribution) of its $K$ neighbors ($x_k$).

(5) Dimensionality reduction: To avoid the *curse of dimensionality* in our high-dimensional dataset, we utilize principal component analysis (PCA), a feature dimensionality reduction technique. PCA is an unsupervised method, where the high-dimensional samples are projected to a lower-dimensional space with minimum loss of information and maximization of the sample variance. The result is a set of *principal components* smaller than the number of features which preserves most of the data variance. The number of components is chosen automatically following the approach in [35].

(6) Classification methods: Finally, we passed the pre-processed data to nine commonly used ML classifiers: extremely randomized trees (ERTs), gradient boosting (GB) classifier, Gaussian process (GP) classifier, KNN, logistic regression (LR) with L1- and L2-regularization, multi-layer perceptron (MLP), random forest (RF), and support vector machines (SVMs). By choosing algorithms of both types (parametric and non-parametric), we aimed to model different aspects of the patient cohort and reduce misclassification in the ensemble learning paradigm. Moreover, some of these classifiers are capable of handling complex aspects of the data, e.g., class imbalances. For instance, classifiers such as LR and KNN can implement a weighted approach to solve the class imbalance problem. Thus, as shown in Table S8, using different classifiers leads to different prediction pipelines. Improved results for the super-learner, as will be shown later, reflect the benefits of implementing many classifiers.

We used the Phase I data to train the TLF prediction component. Hyper-parameter optimization was carried out for each model through an inner cross-validation step. Considering the size of the dataset and the complexity of the models, time requirements for a single training loop with fixed hyper-parameters were not a critical issue. However, hyper-parameter optimization over a broad search space by means of an exhaustive search was infeasible. For that, we followed a two-step approach for hyper-parameter optimization: (1) a low-resolution, low-cost random search over the search space which identified the approximate optimum regions, and (2) an exhaustive grid search limited to the identified regions.

### 4.2. Ensemble Predictions

After parameter tuning, the best fitted models for each of the nine ML models were chosen and are referred to as M1–M9 (cf. Tables S8 and S9 for the best fitted models and the parameter space). Table S8 lists the best selected pre-processing pipelines. In Table S9, all parameters denoted with * represent the best parameters selected in at least one outer cross-validation fold. We used the F1 score as a performance measure and an inner cross-validation technique to decide on the best fit for each classifier. Eventually, we had nine different TLF pipelines, each with a unique setting of pre-processing steps and hyper-parameters for each classification algorithm predicting TLF.

Using a complete outer cross-validation on the entire patient cohort, a decision matrix $M$ was formed (Figure 2), where

$$M[i, j] = P_{\theta_j}(Y_i = 1 | X_i)$$

Here, $X_i$ corresponds to the input data for the $i$-th test patient, $Y_i$ is the predicted class label, and $\theta_j$ are the trained model parameters for the best fitted $M_j$.

From the decision matrix, we derived three ensemble predictions via: (1) unweighted majority voting $M_{mv}$, (2) mean probability $M_{mp}$, and (3) a trained super-learner (SL) $M_{sl}$:

$$M_{mv}[i] = \frac{1}{9}\sum_{j=1}^{9} \mathbb{1}(M[i,j] > 0.5)$$

$$M_{mp}[i] = \frac{1}{9}\sum_{j=1}^{9} M[i,j]$$

$$M_{sl}[i] = \sum_{j=1}^{9} S_b(a_j * M[i,j])$$

where $\mathbb{1}(.)$ is the identity function. The SL combines the individual model predictions via weights $a_j$ learned with a logistic regression to predict the class label, where $S_b$ is the sigmoid function with base $b$.

### 4.3. Risk Score Component

The method developed in the first component is designed to predict the occurrence of future TLF at the time of discharge and is based on the retrospective information available at discharge. In addition to this prediction, we plan to use our method prospectively until the end of the ongoing study. This will allow clinicians to optimize the treatment plans for at-risk patients.

First, using our training dataset (Phase I data), we learn the distribution of TLF over time (Figure 3). To model the time-to-TLF distribution, we use a variety of known probability distributions and find the best fit. Here, we use the Kolmogorov–Smirnov test, a non-parametric test which evaluates the goodness of fit based on the difference between the empirical CDF of the TLF events $F_0(X)$ and the CDF of a candidate probability distribution $F_r(X)$. The null hypothesis $H_0$ is that the data have been sampled from the candidate distribution i.e., $F_0(X) = F_r(X)$. The test statistic $D$ is defined as the maximum difference of two CDFs over the samples $X$:

$$D = \max_x |F_0(X) - F_r(X)| \tag{1}$$

For a good candidate $r^*$, $D$ goes to zero as the sample size $n$ goes to infinity. The critical value for $D$ is found from the Kolmogorov–Smirnov table values for one sample test, which we use to infer the *p*-value. As shown in Figure 3, the best fitted model for risk score is the generalized extreme value distribution (parameters: shape = $-0.24$, loc = 147, scale = 116, and *p*-value = 0.62). Finally, the risk score of a given patient for developing TLF after time $\tau$ is measured as:

$$\text{risk}(\tau) = 1 - F_{r^*}(X = \tau) \tag{2}$$

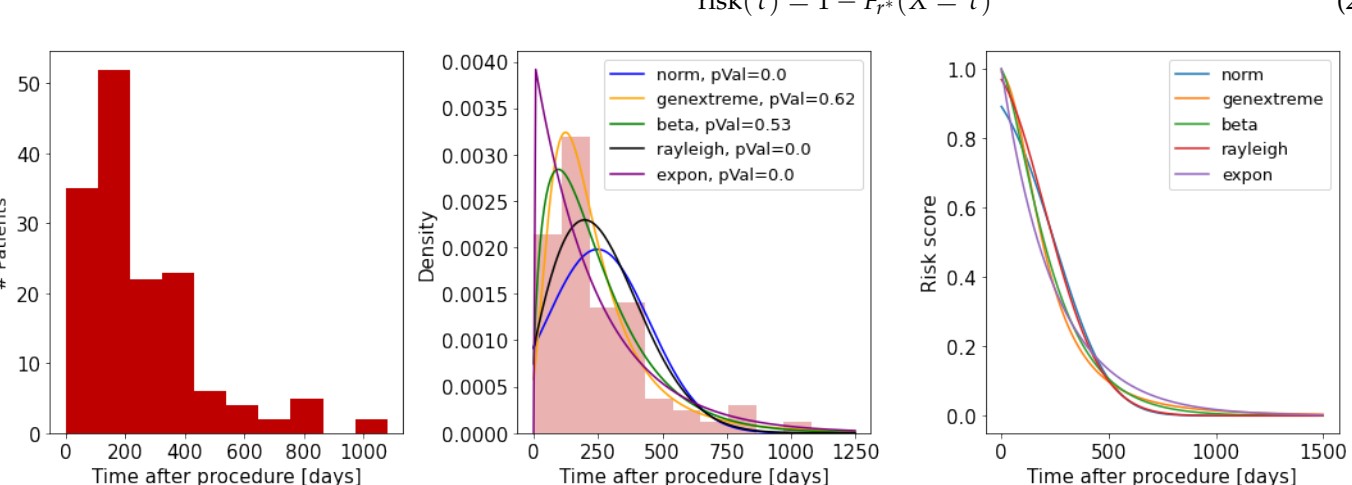

**Figure 3.** Fitted risk score model to TLF distribution: (**left**) histogram of time to TLF; (**middle**) candidate distributions tested against the empirical distribution using the Kolmogorov–Smirnov test; (**right**) the constructed risk score for each distribution.

### 4.4. Update Component

At the time of discharge, using the TLF prediction component, we label a group of patients as at risk. For these patients, at this time point, the risk score model returns a value of 1.0, meaning they have a high chance of developing TLF in the future. When time passes, we can update this risk score, using the time-to-TLF distribution model.

To do so, we updated the risk of TLF for each individual patient at three prospective validation time points: (1) at the end of Phase I (early-term TLF prediction), (2) three months later, at the end of October 2020 (late-term TLF prediction), and (3) at the end of January 2021 (very late-term TLF prediction).

$\tau$ is the time after discharge for each individual patient. If our algorithm predicts TLF for a given patient after discharge, we assign them to the at-risk group and they are followed up more closely. To accommodate resources available at hospitals, we designed a cut-off value for risk. If a patient's risk is above the cut-off value, the patient will be kept in the at-risk group, or otherwise be removed from the group.

## 5. Model Evaluation

We perform two sets of evaluations to report and compare the performance of our model. First, we measure the performance of our models on our cohort using cross-validation, and then we compare it to five state of the art models.

### 5.1. Evaluation of Our Models

We use a nested cross-validation approach to select the best fitted models (inner 4-fold cross-validation) and measure the models' generalization to new and unknown patients (5-fold outer cross-validation). We split patients randomly into 60% training, 20% validation, and 20% test sets while maintaining a similar ratio of TLF patients across folds. During model selection, we also compare combinations of different pre-processing steps using different statistical tests: a Kruskal–Wallis test as an omnibus test and a Wilcoxon sign-ranked test as a paired test.

The performance of each technique is reported using the confusion matrix: true positive (TP), true negative (TN), false positive (FP), and false negative (FN) rates. In addition, we measure precision, recall (TP rate, sensitivity), specificity (TN rate), FP rate and FN rate, and F1 scores. For each of the $F$ cross-validation folds, a confusion matrix $C_f$ ($f = \{1, 2, \ldots, F\}$) of size $2 \times 2$ is computed. Given $F$ confusion matrices, we calculate the mean and standard deviation of the following measures of performance:

$$\text{Precision} = \sum_{f=1}^{F} \frac{\text{TP}_f}{\text{TP}_f + \text{FP}_f}, \ \text{std} = \frac{1}{F-1}\sqrt{\sum_{i=1}^{F}\left(\frac{\text{TP}_f}{\text{TP}_f + \text{FP}_f} - \text{Precision}\right)^2} \quad (3)$$

$$\text{Recall} = \sum_{f=1}^{F} \frac{\text{TP}_f}{\text{TP}_f + \text{FN}_f}, \ \text{std} = \frac{1}{F-1}\sqrt{\sum_{i=1}^{F}\left(\frac{\text{TP}_f}{\text{TP}_f + \text{FN}_f} - \text{Recall}\right)^2} \quad (4)$$

Similarly, we calculate the mean and standard deviation of the following performance metrics:

$$\text{F1} = \frac{\text{TP}}{\text{TP} + \frac{1}{2}(\text{FP} + \text{FN})}$$

$$\text{Specificity} = \text{True Negative Rate} = \frac{\text{TN}}{\text{TN} + \text{FP}}$$

$$\text{False Negative Rate} = \frac{\text{FN}}{\text{TP} + \text{FN}}, \ \text{False Positive Rate} = \frac{\text{FP}}{\text{TN} + \text{FP}}$$

In addition, we also report the receiver operating characteristic (ROC) and precision–recall (PR) curves to compare models. The ROC curve shows the trade-off between the true positive rate and the false positive rate. The PR curve shows the trade-off between precision

and recall for different thresholds and is better suited for imbalanced class distributions. As a baseline, a random classifier, the so-called no-skill classifier, is expected to predict a random class or the same class in all cases.

### 5.2. Evaluation of State of the Art Models

As described in the Background section, we compare our model to five state of the art models: (1) PRESTO-1 [20], (2) PRESTO-2 [11] both used by Sing et al. in 2004, (3) EVENT [21] used by Stolker et al. in 2010, (4) Konigstein [15] used by Konigstein et al. in 2019, and (5) GRACIA-3 [17] used by Sampedro-Gómez et al. in 2020. All these models were designed, trained, and evaluated slightly differently from each other. Here, we discuss how each of the models originally differed from our study and how we provided an equal test bed for the comparison of these models.

In terms of method, PRESTO-1, PRESTO-2, and EVENT use logistic regression (LR) for prediction, Konigstein uses Cox proportional hazard, and GRACIA-3 has used a variety of ML classifiers (such as RF, ERT, LR) and concluded that the ERT classifier was superior.

In terms of features used for training the models, in the original publications, all five models used slightly different feature sets. A summary of features used by these models is provided in Table 3 along with their overlap with our cohort features.

**Table 3.** Features and models used in the original literature for each of the five state of the art models. * = the feature is missing in our dataset and was excluded in the *retrained* experiments.

| Study | Trained Model | Features |
|---|---|---|
| EVENT | LR | Age < 60 years, prior PCI, Left main PCI *, SVG location *, minimum stent diameter $\leq$ 2.5 mm, total stent length $\geq$ 40 mm |
| PRESTO-1 | LR | Lesion length > 20 mm, ACC/AHA type C lesion, previous PCI, treated diabetes mellitus, non-smoker, vessel size (evaluated w.r.t. 3, 3.5, and 5 mm), unstable angina, gender |
| PRESTO-2 | LR | Treated diabetes mellitus, non-smoker, vessel size < 3 mm, length of the lesion (evaluated w.r.t. 10 and 20 mm), ostial length *, previous PCI |
| Konigstein | None | Stent length, moderate/severe calcification, post-procedural diameter stenosis, vessel diameter, hypertension, diabetes, prior coronary artery bypass grafting *, prior PCI |
| GRACIA-3 | ERT | Demographic data (age, weight *, height *, systolic/diastolic blood pressure *, smoking, alcohol consumption *), clinical data (diabetes mellitus, hypertension, dyslipidemia *, family history of cardiovascular diseases *, previous angina, previous MI, previous PCI), medications (ACE/RAA inhibitors *, betablockers *, calcium antagonists *, nitroglycerin *, aspirin *, clopidogrel), angiographic data (vessel disease, drug-eluting stent *, number of implanted stents, tirofiban use *, satisfactory PCI result *, non-reflow *, pre- and post-PCI thrombus/TIMI flow/TMPG */minimal luminal diameter */percent stenosis diameter, percent area stenosis, lesion length), left ventricle function data (end diastolic/systolic volume *, ejection function *), biochemical data (CK, CK-MB, total/LDL cholesterol *, platelets *, leucocytes *, hemoglobin *, hematocrit *, creatinine *) |

In terms of outcomes of interest, PRESTO-1, PRESTO-2, and GRACIA-3 were designed for prediction of stent restenosis (SR). EVENT was designed for prediction of target lesion stent restenosis (TLR), and Konigstein was designed for TLF. These three outcomes are clinically closely related to each other. Our definition of TLF in this paper includes both TLR and SR.

In terms of reporting model parameters, PRESTO-1, PRESTO-2, and EVENT report the LR model parameters ($\beta$-coefficients) after training on their own original database. Konigstein provides the hazard ratio (HR) for a number of features, but they were not directly used for cross-validation on held-out cohorts. GRACIA-3 does not provide the trained ERT model parameters.

To compare these models to ours, we performed two sets of experiments: (1) in this scenario, the models come pre-trained from the original literature and are only tested on our database. To do so, we employ the already trained models and their parameter values (reported in their original literature), and test their prediction power on our database. We refer to this set of experiments as *test-only*, and (2) we retrain these models on our database as well, which leads to the generation of new parameter values. Using the new values, we then test the models' prediction power on our database. We do so using the same 5-fold cross-validation technique as above. We refer to this set of experiments as *retrained*.

Since each model was originally designed for slightly different clinical outcomes of interest (TLF, TLR, or SR), and to provide a fair comparison between techniques, we repeat both sets of experiments once for predicting TLF (the focus of Konigstein's study and ours) and once for TLR (the focus of the PRESTO-1, PRESTO-2, EVENT, and GRACIA-3 studies).

Similar to evaluation of our own model, we report the area under the ROC curve as well as the precision of all models (for a fixed sensitivity of 90%). The validation results are reported and discussed in the next section.

## 6. Results and Discussion

Following the described evaluation design, we now describe the results and findings of each evaluation set.

### 6.1. Performance of Our Models

Table 4 shows the performance of different components of our model which are validated both retrospectively and prospectively. The rows indicate the nine individual ML models, the three ensemble learners, and the prospective validation for prediction of early-, late-, and very late-term TLF. The nine individual models are: extremely randomized trees (ERTs), gradient boosting (GB), Gaussian process (GP), k-nearest neighbor (KNN), logistic regression with L1- and L2-regularization (L1-LR, L2-LR), multi-layer perceptron (MLP), random forest (RF), and support vector machines (SVMs). The three ensemble classifiers are: majority voting, mean probability, and super-learner.

The results show that due to the nature of imbalanced class distributions, there is a steep trade-off between higher TP and TN. For example, KNN has the highest TPR (91%) but with very low TNR (21%). This means that KNN correctly predicts 91% of the TLF patients, but flags 79% of the healthy patients as at risk (FP = 1433), requiring clinicians to pay attention to a large cohort of patients. On the other hand, the RF classifier finds almost all (TNR = 0.94%) healthy patients correctly (TN = 1707) and generates the smallest at-risk cohort (FP = 117) to reduce clinicians' workload. Consequently, it misclassifies the majority of TLF patients (FNR = 89%). Looking at both ROC and PR curves (Figure 4), between the nine basic classifiers, GP performs best with the largest area under the curve (AUC). To statistically evaluate the performance of these classifiers, we applied an F-test. The null hypothesis was defined as 'there are no differences between the classification performances'. We first computed the F-statistic, then obtained the *p*-value from a cumulative F-distribution function. We rejected the null hypothesis with a *p*-value of <0.001 and an F-statistic of 585. The results showed we could not observe significant differences between the SL model and the other nine models for TLF prediction.

In practice however, clinicians are interested in models with higher TPR while sacrificing moderate resources for FP patients. In this case, L2-LR with TPR = 54% and TNR = 62% is a better choice, generating 698 at-risk patients (FP = 698).

We trained the SL only once at the end of Phase I to predict TLF at discharge. To analyze the performance of our method over time, we performed the first prospective TLF

prediction analysis at the end of Phase I (31 July 2020). At this time point, our algorithm had flagged 705 patients as at risk, projecting a huge workload for the clinicians. To reduce the number of at-risk patients at this time point, we used the risk score model to generate a risk score for each patient, estimating their chance of TLF at that time point.

**Table 4.** Performance of the nine individual ML models in our study, three ensemble learners, and prospective validations for prediction of early-, late-, and very late-term TLF. * = calculated on sum of confusion matrices. NA = AUC-ROC is not measurable for these two-step follow-up models.

| Model | TNR Specificity | FPR | FNR | TPR Recall Sensitivity | Precision | F1 | AUC-ROC |
|---|---|---|---|---|---|---|---|
| ERT | $0.81 \pm 0.02$ | $0.19 \pm 0.02$ | $0.66 \pm 0.07$ | $0.34 \pm 0.07$ | $0.13 \pm 0.02$ | $0.18 \pm 0.03$ | $0.62 \pm 0.01$ |
| GMB | $0.9 \pm 0.02$ | $0.1 \pm 0.02$ | $0.82 \pm 0.06$ | $0.18 \pm 0.06$ | $0.12 \pm 0.02$ | $0.14 \pm 0.03$ | $0.62 \pm 0.01$ |
| GP | $0.77 \pm 0.03$ | $0.23 \pm 0.03$ | $0.6 \pm 0.07$ | $0.4 \pm 0.07$ | $0.12 \pm 0.01$ | $0.19 \pm 0.02$ | $0.63 \pm 0.01$ |
| KNN | $0.21 \pm 0.02$ | $0.79 \pm 0.02$ | $0.09 \pm 0.04$ | $0.91 \pm 0.04$ | $0.09 \pm 0.01$ | $0.16 \pm 0.01$ | $0.62 \pm 0.01$ |
| L1-LR | $0.86 \pm 0.03$ | $0.14 \pm 0.03$ | $0.76 \pm 0.03$ | $0.24 \pm 0.03$ | $0.13 \pm 0.02$ | $0.17 \pm 0.02$ | $0.62 \pm 0.01$ |
| L2-LR | $0.62 \pm 0.04$ | $0.38 \pm 0.04$ | $0.48 \pm 0.05$ | $0.52 \pm 0.05$ | $0.1 \pm 0.01$ | $0.17 \pm 0.01$ | $0.63 \pm 0.01$ |
| MLP | $0.82 \pm 0.08$ | $0.18 \pm 0.08$ | $0.69 \pm 0.1$ | $0.31 \pm 0.1$ | $0.13 \pm 0.02$ | $0.18 \pm 0.02$ | $0.62 \pm 0.01$ |
| RF | $0.94 \pm 0.02$ | $0.06 \pm 0.02$ | $0.89 \pm 0.03$ | $0.11 \pm 0.03$ | $0.13 \pm 0.04$ | $0.11 \pm 0.03$ | $0.63 \pm 0.01$ |
| SVM | $0.76 \pm 0.05$ | $0.24 \pm 0.05$ | $0.65 \pm 0.08$ | $0.35 \pm 0.08$ | $0.11 \pm 0.01$ | $0.16 \pm 0.01$ | $0.62 \pm 0.01$ |
| Majority voting * | 0.81 | 0.19 | 0.66 | 0.34 | 0.13 | 0.19 | $0.62 \pm 0.01$ |
| Mean probability * | 0.80 | 0.20 | 0.66 | 0.34 | 0.12 | 0.18 | $0.63 \pm 0.01$ |
| SL * (early-term) | 0.61 | 0.39 | 0.46 | 0.54 | 0.10 | 0.17 | $0.62 \pm 0.01$ |
| SL * (late-term) | 0.87 | 0.13 | 0.47 | 0.53 | 0.27 | 0.36 | NA |
| SL * (very late-term) | 0.92 | 0.08 | 0.47 | 0.53 | 0.36 | 0.43 | NA |

We also chose an empirical cut-off value to reduce the number of at-risk patients to one-third (0.33).Therefore, we kept 234 patients in the at-risk group and moved the remaining 471 patients to the not-at-risk group. At this point, we achieved a performance with TNR = 61.35% and TPR = 54.30% (Figure 5).

The second prospective validation point was three months later (31 October 2020). During these three months, eight patients had developed TLF for the first time (right-censored/healthy during the July cohort). Our algorithm successfully found two of them, even though they were right-censored in July and were used as TLF-negative samples for training the models. The other six underdiagnosed patients were correctly classified as not at risk for the end of July. We, however, counted these six patients as FN during the October validation. For the remaining 232 still at-risk patients, we updated their risk score to accommodate for the 90 days that had passed since the last calculation of the risk score in July. Passing through the risk score cut-off, at the end of October, our algorithm flagged 155 patients as still at risk and moved 77 to the not-at-risk group. At this point, we achieved a performance of TNR = 87.22% and TPR = 52.83%.

The third prospective validation point was three months later (31 January 2021). During these three months, four additional patients had developed TLF for the first time. Our algorithm successfully found two of them, even though they were right-censored both in July and October and were used as TLF-negative samples for training the models. The other two underdiagnosed patients were in our not-at-risk group by the end of July. Similarly, we counted these two patients as FN during January validation. For the remaining 153 still at-risk patients, we updated their risk score to accommodate for the 90 days that had passed since the last calculation of the risk score in October. Passing through the same risk

score cut-off, at the end of January, our algorithm flagged 105 patients as still at risk and moved 48 to the not-at-risk group. By the time of the submission of this paper, no new patients were reported with TLF. This put our potential FPR at 8.44%. At this point, we achieved a performance of TNR = 91.66% and TPR = 52.76%.

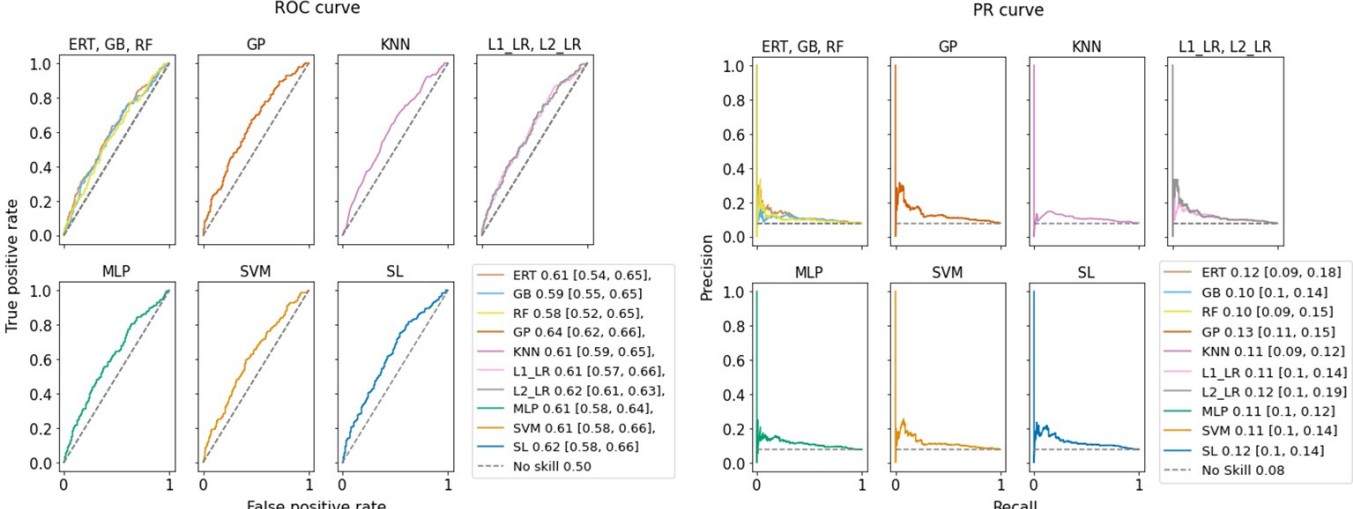

**Figure 4.** Prediction performance of our ML models. Values shown in legends are areas under the curve and their 95% confidence interval for (**left**) ROC curve and (**right**) precision–recall (PR) curve. F-statistics show no significant difference between predictions.

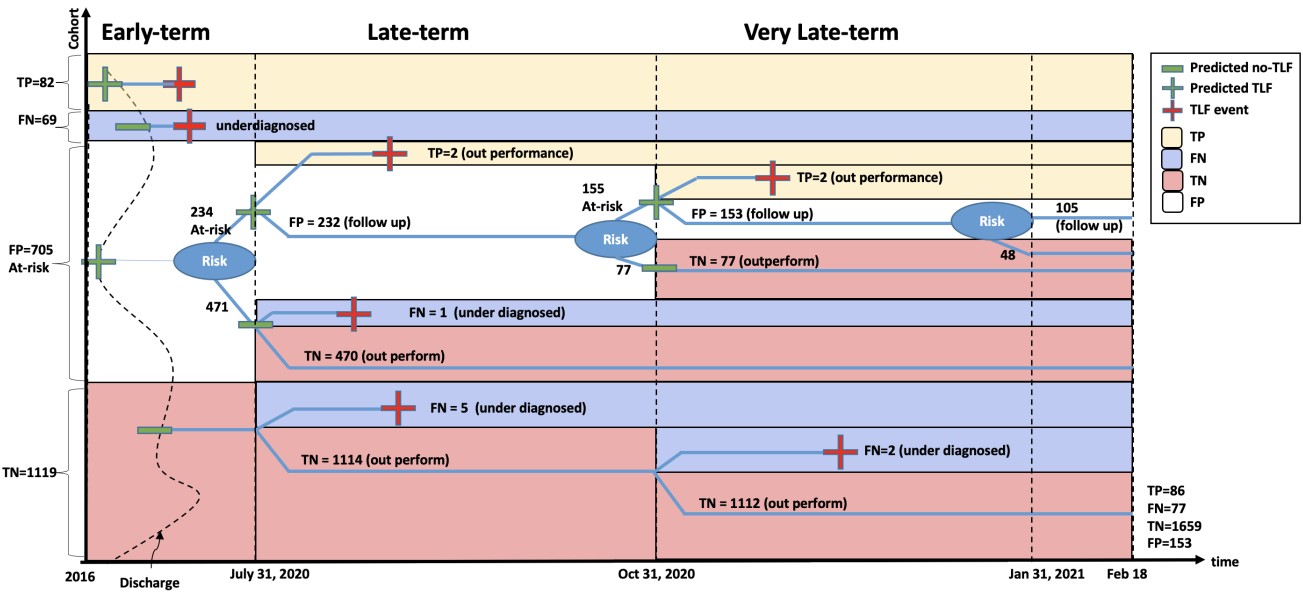

**Figure 5.** Retrospective and prospective validation of our TLF prediction model.

## 6.2. Performance of the State of the Art Models

To perform a fair comparison between our model (SL early-term) and five state of the art models (PRESTO-1, PRESTO-2, EVENT, Konigstein, and GRACIA-3), we designed two sets of experiments: (1) *test-only* using the pre-trained model on our database only for testing, and (2) *retrained* where all models are retrained and tested on our database using a 5-fold cross-validation approach. We repeated both sets of experiments to predict both TLR and TLF, separately. By doing so, we can demonstrate the performance of all models regardless of the particularity of their cohorts, features, and outcome of interest.

Table 5 presents the list of model parameter values ($\beta$-coefficients for the three logistic regression-based models) as were reported in the literature by the respective authors (*test-only*), and after retraining on our database (*retrained*). The outcome of interest here is TLR, similar to the original study. As shown, EVENT has only one parameter overlap with the two PRESTO studies. When a given parameter did not exist in our database, we denoted it with NA, which appears only three times (two variables of EVENT and one variable of PRESTO-2). Hence, our database can be used to meaningfully compare these models. For the *test-only* experiment, when a variable is not available in our database (NA), we impute it by sampling from a Bernoulli distribution with a success probability equal to 50%.

Table 5 shows that $\beta$-coefficients are different between *test-only* and *retrained* experiments. Since the classifier is the same between these three models, we argue that the differences among the trained $\beta$-coefficients are most probably caused by the difference in cohort distributions and characteristics. For instance, if an outcome and a feature are confounded by a hidden variable whose distribution is different among the cohorts, the corresponding coefficient of that feature would differ between cohorts. For the same reason, it is important to note that these coefficients must be interpreted with care, especially when the goal is to find causal relations between the features and the outcome.

For the retrained LR model, we also report the model intercept (Table 5) which has not been reported in other studies. The term intercept in an LR model gives the prior odds of the outcome when all of the features have zero values, or are missing. The intercept can be interpreted as a characteristic of the cohort. A high intercept value hints toward a bias in the model preferring to assign a positive prediction to patients with many missing features.

**Table 5.** The model parameter values ($\beta$-coefficients for logistic regression-based models) as were reported in the literature by authors (*test-only* columns), and after retraining on our database (*retrained* columns); PRESTO-1 [20], PRESTO-2 [11], EVENT [21]. The outcome of interest here is TLR, similar to the original studies. The last row indicates the logistic regression model intercept, only reported for retrained models, and not reported by the authors. NA means the parameter did not exist in our database; - means the parameter did not exist in the original model.

| Variables | EVENT | | PRESTO-1 | | PRESTO-2 | |
| --- | --- | --- | --- | --- | --- | --- |
| | *Test-Only* | *Retrained* | *Test-Only* | *Retrained* | *Test-Only* | *Retrained* |
| Patient age < 60 y | 0.401 | 0.143 | - | - | - | - |
| Left main PCI | 1.144 | NA | - | - | - | - |
| SVG location | 0.876 | NA | - | - | - | - |
| Minimum stent diameter $\leq$ 2.5 mm | 0.430 | 0.175 | - | - | - | - |
| Total stent length $\geq$ 40 mm | 0.577 | 0.911 | - | - | - | - |
| Prior (previous) PCI | 0.604 | −0.075 | 0.344 | 0.048 | - | - |
| ACC/AHA type C lesion | - | - | 0.593 | −0.011 | - | - |
| Treated diabetes mellitus | - | - | 0.344 | 0.146 | 0.372 | 0.241 |
| Unstable angina | - | - | 0.174 | 0.327 | - | - |
| Female gender | - | - | 0.140 | −0.460 | - | - |
| Non-smoker | - | - | 0.329 | −0.278 | 0.293 | 0.493 |
| Ostial length | - | - | - | - | 0.600 | NA |
| Lesion length $\geq$ 20 mm | - | - | 0.728 | −0.005 | 0.859 | 0.050 |
| Vessel size | | | | | | |
| $\leq$3 mm | - | - | 0.565 | 0.321 | 0.278 | 0.014 |
| 3–3.5 mm | - | - | 0.365 | 0.266 | - | - |
| 3.5–4 mm | - | - | 0.166 | 0.0178 | - | - |
| >4 mm | - | - | 0.000 | 0.000 | - | - |
| Model intercept | - | 0.093 | - | 0.048 | - | −0.580 |

We have reported the performance (AUC-ROC and precision score corresponding to 90% recall score) of these models for predicting TLF and TLR in Table 6. The results under the *test-only* and the *retrained* columns have been generated by using the original model

parameters or by retraining the model on our dataset, respectively. Since the articles about the Konigstein and GRACIA-3 models do not provide direct pre-trained models, we have denoted them as NA in the *test-only* results. In addition, the original Konigstein model only includes hazard ratio values for a set of variables and does not include a predictive model. To report *retrained* for Konigstein, we used their set of features, trained multiple predictive models (with nine conventional techniques), and reported the performance of the best model (random forest).

**Table 6.** Comparison of performance between our model (SL early-term), and five of the state of the art models. Reported values are AUC-ROC and their standard deviation, as well as precision score for the model when it was set to achieve 90% sensitivity. NA means models do not exist.

| Model | | TLR Prediction | | TLF Prediction | |
|---|---|---|---|---|---|
| | | *Test-Only* | *Retrain* | *Test-Only* | *Retrain* |
| EVENT | AUC-ROC | $0.51 \pm 0.01$ | $0.54 \pm 0.06$ | $0.51 \pm 0.01$ | $0.54 \pm 0.06$ |
| | Precision | 0.062 | 0.066 | 0.076 | 0.081 |
| PRESTO-1 | AUC-ROC | $0.51 \pm 0.01$ | $0.55 \pm 0.05$ | $0.51 \pm 0.01$ | $0.56 \pm 0.03$ |
| | Precision | 0.065 | 0.070 | 0.079 | 0.083 |
| PRESTO-2 | AUC-ROC | $0.52 \pm 0.01$ | $0.54 \pm 0.07$ | $0.51 \pm 0.01$ | $0.52 \pm 0.02$ |
| | Precision | 0.063 | 0.066 | 0.077 | 0.077 |
| Konigstein | AUC-ROC | NA | $0.54 \pm 0.07$ | NA | $0.49 \pm 0.06$ |
| | Precision | | 0.064 | | 0.074 |
| GRACIA-3 | AUC-ROC | NA | $0.61 \pm 0.06$ | NA | $0.58 \pm 0.03$ |
| | Precision | | 0.072 | | 0.091 |
| SL (early-term) | AUC-ROC | $0.64 \pm 0.02$ | $0.62 \pm 0.01$ | NA | $0.62 \pm 0.01$ |
| | Precision | 0.067 | 0.077 | | 0.091 |

As shown in Table 6, the performances of all models slightly improved from *test-only* to *retrained*, reflecting the fact that models pre-trained on original cohorts might not give the best results when tested on a different cohort. Furthermore, EVENT, PRESTO-1, PRESTO-2, Konigstein, and GRACIA-3 models did not reach the same level of performance in the *retrained* step as reported in their respective original studies which again points to the difference among the collected features between cohorts. Our SL model outperform all other state of the art models. However, GRACIA-3 has again outperformed the other four models both in TLF and TLR prediction, similar to the report in their original paper. This can be explained by larger number of features that was used in this model (Table 3). For the same reason, GRACIA-3 obtained the closest level of performance to our SL model. The performances for TLR and TLF predictions among all of the models are close to each other, reflecting the high degree of correlation between these two outcomes. Nevertheless, it is surprising that among the first three models, the *test-only* precision scores for TLF are higher than those of TLR, even though the models have been proposed for predicting SR and TLR outcomes. This difference might not be statistically significant and simply due to variability in our database.

As a summary, the better performance of our model compared to other models was expected, as our database contains more features and we designed a more detailed model.

## 7. Conclusions

In this paper, we presented a machine learning (ML) model to predict TLF using information available before patient discharge. We used our BIOSOLVE-IV post-market study dataset (September 2016–2025) including data of 1975 patients located across the world. These patients had received the Magmaris sirolimus-eluting bioresorbable stent (Biotronik AG, Bülach, Switzerland). For each patient, a total of 86 variables were measured, composed of information before their surgery until after discharge. Additionally, regular follow-up information for up to five years after discharge was collected.

Our model consists of three components: (1) TLF prediction at discharge via combination of nine ML models and a super-learner, (2) prediction of a risk score for time to TLF, and (3) updating the risk score for at-risk patients during their follow-ups. We train and evaluate our models, both retrospectively and prospectively, at three times points: discharge, end of Phase I (early-term prediction), three months after Phase I (late-term prediction), and six months after Phase I (very late-term prediction).

The five main challenges of this problem are: (1) it is clinically extremely difficult to predict TLF, (2) it is important to optimize between the tool's sensitivity and the size of the at-risk cohort, (3) unbalanced class sizes hamper training, (4) missingness in the outcome introduces bias in inference, and (5) cohort selection bias exists due to restrictive inclusion criteria in our clinical study.

Our main contributions in this paper are: (1) the development of a novel ML method for the identification of patients at risk of TLF. This model can predict the onset of TLF at any time point after discharge from hospital. To the best of our knowledge, such an analysis on TLF is the first of its kind. (2) Training and validation are performed with our international, multi-site cohort, with comprehensive variables collected during treatment and 5 years of frequent follow-ups. Data were collected from 120 medical centers in over 25 countries across the globe. (3) We evaluate our model against five state of the art models via multiple experiments. (4) We demonstrate a successful retrospective and prospective evaluation of our model in three time frames (early-, late-, and very late-term) while maintaining a reasonable at-risk cohort size.

Our algorithm successfully predicts TLF for 86 patients (TPR = 52.76%) (82 in early term, two in late term, and two in very late term). It identifies 1635 patients as not at risk (TNR including right-censored patients = 90.23%). However, our model underdiagnoses a total of 77 patients (FNR = 47.23%) (69 in early term, six in late term, and two in very late term). Even though our algorithm was trained only on Phase I data, it successfully identifies four (33%) of the right-censored TLF patients. The results also show that the risk score component helps to reduce the number of follow-up patients to 33% (thus saving clinicians' effort) while maintaining the very late-term prediction performance.

Additionally, we compared our choice of ML classifier and features against five state of the art studies for predicting TLF and TLR outcomes. The comparison was carried out via two sets of test-only and retrain experiments, where the results of other models (if applicable) were compared to that of ours when we reused their original classifier parameters, and when we retrained their proposed classifier on our dataset, respectively. The test-only results were generally lower, which suggests that trained models on other datasets might not give the best result for a new dataset due to differences in the study population. Our model also achieved better results in the retrain step compared to other models. This higher performance might be due to our comprehensive feature set, and combining the power of multiple classifiers in the SL model.

To understand how our model will perform when fewer features are available, we can use the results in the comparison table (Table 6). The two retrained models for Konigstein and GARCIA-3 provide an approximate answer, as they generally use the same conventional ML models (without combining them) on a much smaller feature set.

To improve the prediction power, we recommend including a routine carotid CT angiography (CCTA) during the first follow-up visit. Evidence from the literature [5–8] has shown that radiomics features extracted from CCTA, even though costly, have improved prediction power for cardiac outcomes. These analyses can help in measuring significant biomarkers such as carotid artery luminal stenosis, calcification of the artery walls, the napkin-ring sign, epicardial adipose tissue, and periconary adipose tissue, hence improving our clinical understanding for patient recovery trajectories.

**Supplementary Materials:** The following are available online at https://www.mdpi.com/article/10.3390/app11156986/s1, Figure S1: Overview of BIOSOLVE-IV post-market study, Figure S2: The missingness in the BIOSOLVE-IV study is low with a missingness rate <0.2 for 75.0% of the patients and <0.1 for 85.0% of the features, Table S1: Overview of significant predictors of cardiovascular

disease in literature, Table S2: Cohort features (part 1): patient characteristics and statistics, Table S3: Cohort features (part 2): procedural information and statistics, Table S4: Cohort features (part 3): lesion characteristics and statistics, Table S5: Cohort features (part 4): medication and statistics, Table S6: Cohort features (part 5): discharge information and statistics, Table S7: Explanation of ordinal features, Table S8: Model structure for final TLF models, Table S9: Fixed and tuned hyper-parameters for TLF models.

**Author Contributions:** Data curation, C.B.; Formal analysis, E.P.; Investigation, M.S. and C.B.; Methodology, A.Z.; Supervision, N.A.; Validation, E.P. and A.Z.; Visualization, E.P. and N.A.; Writing—original draft, E.P.; Writing—review and editing, A.Z. and N.A. All authors have read and agreed to the published version of the manuscript.

**Funding:** This research received no external funding.

**Institutional Review Board Statement:** The study was conducted according to the guidelines of the Declaration of Helsinki. This international study was approved by the respective committees and competent authorities, where applicable, of the participating sites. All the patients have provided written informed consent.

**Informed Consent Statement:** Informed consent was obtained from all subjects involved in the study.

**Data Availability Statement:** Data cannot be shared publicly due to a confidentiality agreement. However, access to data for research purposes is possible for eligible institutes according to GDPR rules.

**Acknowledgments:** The authors would like to thank Theresa Wirth, Henrik von Kleist, Hugo Loureiro, and Philipp Stolka for careful review of the manuscript and their constructive feedback.

**Conflicts of Interest:** These authors declare no conflict of interest: N.A., E.P., A.Z. The following authors are employees of Biotronik, the company which has provided the data for analysis: M.S. and C.B. Theses authors had no role in study design, data collection, and analysis. They were involved in the decision to publish, and reviewed the article before submission.

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
