# Peer review of "Early-, Late-, and Very Late-Term Prediction of Target Lesion Failure in Coronary Artery Stent Patients: An International Multi-Site Study"

_applsci, doi:10.3390/app11156986_

Round 1
Reviewer 1 Report
Considering that the typical intervention for coronary artery disease is stent implantation, this manuscript addressed the prediction of intervention success by using post-discharge adverse outcomes, also known as target lesion failure (TLF).
The TLF prediction was performed by combining nine conventional models and a super-learner. In addition, a risk score predictor for time-to-TLF, as well as an update function to manage the size of the at-risk cohort, were proposed.
The method is interesting and the results are reasonable, although statistical validation among the investigated machine learning models would be beneficial. However, the literature background might be extended. The manuscript is well prepared but further proofreading would be beneficial to increase the quality of the manuscript.
My main comments are listed in what follows.
1) Abstract: conclusive remarks should be extended.
2) Keywords: more specific terms should be provided.
3) Section 1: the structure of the manuscript should be provided at the end of the Introduction.
4) Section 2: the used features should be better summarized and described in the main text along with the Supplementary Tables.
5) The importance of imaging for prediction, Computed Tomography (CT) in particular, should be introduced. Especially, in terms of pericoronary adipose tissue
[Lin A., Kolossváry M., Yuvaraj J., et al. (2020) Myocardial infarction associates with a distinct pericoronary adipose tissue radiomic phenotype: a prospective case-control study. JACC: Cardiovascular Imaging, 13(11), 2371-2383. DOI: 10.1016/j.jcmg.2020.06.033] and epicardial fat [Militello C., Rundo L., Toia P., et al. (2019) A semi-automatic approach for epicardial adipose tissue segmentation and quantification on cardiac CT scans. Computers in Biology and Medicine, 114, 103424. DOI: 10.1016/j.compbiomed.2019.103424].
Please consider to introduce and discuss these very recent articles.
6) Concerning modelling, radiomics applications should be introduced for both coronary and carotid arteries:
- Kolossváry, M., Karády, J., Szilveszter, B., Kitslaar, P., Hoffmann, U., Merkely, B., & Maurovich-Horvat, P. (2017). Radiomic features are superior to conventional quantitative computed tomographic metrics to identify coronary plaques with napkin-ring sign. Circulation: Cardiovascular Imaging, 10(12), e006843. DOI: 10.1161/CIRCIMAGING.117.006843
- Le, E. P., Rundo, L., Tarkin, J. M., Evans, N. R., Chowdhury, M. M., Coughlin, P. A., ... & Rudd, J. H. (2021). Assessing robustness of carotid artery CT angiography radiomics in the identification of culprit lesions in cerebrovascular events. Scientific reports, 11, 3499. DOI: 10.1038/s41598-021-82760-w
Please discuss the potential of radiomics with the support of these relevant articles.
7) Figure 3 might be improved since all the ROC curve lines are almost overlapped. No statistical validation was presented.
8) Section 5: conclusive remarks need to be substantially extended, as well as a feasible plan for future work should be provided.
Author Response
Please see attached PDF document.

Reviewer 2 Report
Тhe proposed paper is devoted to the presentation of a novel machine-learning model used for prediction of target lesion failure (TLF) in coronary artery stent patients taking into account information available before patients discharge as well as follow-up information for up to five years after discharge. The model can predict the onset of TLF at any time point after discharge from hospital. The presented results show that the proposed model gives satisfactory results while maintaining a reasonable at-risk cohort size.
The presentation of the main results is clear and comprehensive. The results are valuable and worthy of being published taking into account their possible applications in clinics for optimization of treatment plans for at-risk patients.
Minor revisions are suggested to improve the quality of the exposition:
p. 6, line 217: It should be “Eventually” instead of “Eventuallyw”
p. 7, line 241: I suggest to write “distribution i.e.” instead of “distribution, i.e.,”
p. 13, line 450: The names of the authors of Ref. 16 should be corrected
Reviewer 3 Report
- Please highlight the main innovations of the paper.
- The introduction lacks more detail on the proposed approach, especially its highlights compared to the literature.
- The introduction does not clearly present the motivation and the document contents.
- I advise a better organization of the literature review section. For example, a chart with simple explanations of the techniques could be provided to the reader.
- The final paragraph for organizing the paper is missing.
- The proposed methods must be compared with another method to confirm the correctness of obtained results as well as the strengths and limits of the proposed approach.
- In average, relatively obsolete citation sources are used. A higher number of newest and more representative sources must be used.
- The language quality must be improved. There are many typing errors, which could be detected by a spellchecker.
- Part of material and method section has to be explained better paying more attention to the equations description so that the paper could be understood by inexperienced readers, too.
- For as concern the results section and so the results discussion, i found it very poor in content.
- I have many concerns about the technical requirements of the testbed environment. I think the model is dependent on very expensive resources and this is a drawback. The authors should point out how can the model reduces the high overhead introduced by their methodology.
- The publication contains hints towards a scientific contribution that is worthwhile to publish. The areas of publication mentioned should be made more detailed, precise, and streamlined, and the central theme should be polished.
- Due to the lack of detail in the selection of samples from the dataset, it is unclear whether there is a bias in the samples, selected or accidental, which in turn could affect the results. However, due to the lack of traceability, the strong results cannot unfold and fall behind.
- In its current form the publication describes the pre-processing and feature extraction in detail, also mathematically, but remains unclear in the techniques in between and for final classification.
- The publication lacks a contribution statement, which hinders the reader to distinguish between the current state of the art and the contributions that the authors add to the field. Moreover, the authors did not clarify / specify of the contribution in the text any further.
- In order to regain space, I highly suggest removing the descriptions of the process or only refer to it shortened. Researchers in the field know the definitions. Instead, refer to the points of the proposed method directly and how it is implemented.
- I strongly suggest a figure, that show the global interaction of each system, as well as a declaration, what step is upfront (pre-processing), during training and deployment / operation of such system.
- Experiments are weak. There is no comparison with other approaches in the literature.
- The methodology is extensive. The authors could use graphical resources to help understand the steps to be performed by the approach.
Round 2
Reviewer 1 Report
The Authors have mostly addressed the issues raised in the first revision round. The manuscript is now in better shape. The Reply Letter is exhaustive and the changes have been marked in the manuscript.
The Figures have been revised and the English language has been carefully proofread.
However, there are additional improvements that should be taken into account by the Authors.
My suggestions are listed in what follows.
* Section 1 is too long and the actual contributions appear to be diluted.
I would suggest avoiding Table 1 and Figure 1 in the Introduction and move them, along with the related text, to a new Section 2 "Background" for better introducing the context. This will help the reader to identify the main achievements presented in this paper.
* Reference [8] repeats also reference [7]. Please revise as follows:
"8. Le EP, Rundo L, Tarkin JM, Evans NR, Chowdhury MM, Coughlin PA, Pavey H, Wall C, Zaccagna F, Gallagher FA, Huang Y. Assessing robustness of carotid artery CT angiography radiomics in the identification of culprit lesions in cerebrovascular events. Scientific reports. 2021 Feb 10;11(1):1-4. Militello C, Rundo L, Toia P, Conti V, Russo G, Filorizzo C, Maffei E, Cademartiri F, La Grutta L, Midiri M, Vitabile S. A semi-automatic approach for epicardial adipose tissue segmentation and quantification on cardiac CT scans. Computers in biology and medicine. 2019 Nov 1;114:103424"
Reviewer 3 Report
Based on the comments raised by the examiners all of the doubtful and blurred areas have been well explained by the authors. Now it is ready for publication.
